# Beneficial Effect of H_2_S-Releasing Molecules in an In Vitro Model of Sarcopenia: Relevance of Glucoraphanin

**DOI:** 10.3390/ijms23115955

**Published:** 2022-05-25

**Authors:** Laura Micheli, Emma Mitidieri, Carlotta Turnaturi, Domenico Vanacore, Clara Ciampi, Elena Lucarini, Giuseppe Cirino, Carla Ghelardini, Raffaella Sorrentino, Lorenzo Di Cesare Mannelli, Roberta d’Emmanuele di Villa Bianca

**Affiliations:** 1Department of Neuroscience, Psychology, Drug Research and Child Health-Neurofarba—Section of Pharmacology and Toxicology, University of Florence, 50139 Florence, Italy; laura.micheli@unifi.it (L.M.); clara.ciampi@stud.unifi.it (C.C.); elena.lucarini@unifi.it (E.L.); carla.ghelardini@unifi.it (C.G.); 2Department of Pharmacy, School of Medicine and Surgery, University of Naples Federico II, 80131 Naples, Italy; emma.mitidieri@unina.it (E.M.); carlotta.turnaturi@unina.it (C.T.); domenico.vanacore@unina.it (D.V.); cirino@unina.it (G.C.); demmanue@unina.it (R.d.d.V.B.); 3Department of Molecular Medicine and Medical Biotechnology, School of Medicine, University of Naples Federico II, 80131 Naples, Italy; raffaella.sorrentino@unina.it

**Keywords:** sarcopenia, skeletal muscle, glucoraphanin, hydrogen sulfide, hydrogen-sulfide-releasing molecules, cystathionine-γ-lyase, cystathionine-β-synthase

## Abstract

Sarcopenia is a gradual and generalized skeletal muscle (SKM) syndrome, characterized by the impairment of muscle components and functionality. Hydrogen sulfide (H_2_S), endogenously formed within the body from the activity of cystathionine-γ-lyase (CSE), cystathionine- β-synthase (CBS), and mercaptopyruvate sulfurtransferase, is involved in SKM function. Here, in an in vitro model of sarcopenia based on damage induced by dexamethasone (DEX, 1 μM, 48 h treatment) in C2C12-derived myotubes, we investigated the protective potential of exogenous and endogenous sources of H_2_S, i.e., glucoraphanin (30 μM), L-cysteine (150 μM), and 3-mercaptopyruvate (150 μM). DEX impaired the H_2_S signalling in terms of a reduction in CBS and CSE expression and H_2_S biosynthesis. Glucoraphanin and 3-mercaptopyruvate but not L-cysteine prevented the apoptotic process induced by DEX. In parallel, the H_2_S-releasing molecules reduced the oxidative unbalance evoked by DEX, reducing catalase activity, O_2_^−^ levels, and protein carbonylation. Glucoraphanin, 3-mercaptopyruvate, and L-cysteine avoided the changes in myotubes morphology and morphometrics after DEX treatment. In conclusion, in an in vitro model of sarcopenia, an impairment in CBS/CSE/H_2_S signalling occurs, whereas glucoraphanin, a natural H_2_S-releasing molecule, appears more effective for preventing the SKM damage. Therefore, glucoraphanin supplementation could be an innovative therapeutic approach in the management of sarcopenia.

## 1. Introduction

Sarcopenia is a skeletal muscle (SKM) disorder characterized by a gradual and generalized impairment in terms of structure and functionality, frequently associated with physical handicap, significantly worsening patient’s quality of life and, in worst cases, leading to death [1,2]. In simpler terms, it could be best defined as SKM insufficiency or failure [3]. Sarcopenia is a complex multifactorial condition [4,5], though aging is the major underlying cause [5]. Age-related sarcopenia can be enhanced by various factors, including sedentary lifestyle, poor nutrition, chronic illness, and disturbance in the peripheral and/or central nervous systems. However, even if sarcopenia is usually considered an elderly disorder, its development can also occur in young people and in particular in those suffering from autoimmune diseases [6]. In addition, hormonal changes contribute to the decrease in muscle mass and performance; indeed, although the involvement of several hormones such as insulin, sexual hormones, corticosteroids, thyroid hormones, growth hormone, prolactin, and catecholamines are described in the onset and development of sarcopenia, it is still not clear how they, respectively, specifically act on SKM [4]. Additionally, genetic susceptibility plays a role, evidence that can justify the individual and group variability in the incidence rate of sarcopenia [7,8]. Finally, it should be underlined that the use of drugs such as glucocorticoids and chemotherapeutics also participates in the development of sarcopenia [5,9,10]. Therefore, numerous factors are involved in the onset and development of sarcopenia.

In spite of its clinical relevance, sarcopenia is still a poorly recognized and therefore poorly managed syndrome in routine clinical practice. Certainly, an early diagnosis and an appropriate therapeutic approach are crucial to improving outcomes in patients with sarcopenia. The lack of a proper physical activity is related to the reduction in muscle mass and performance, so a good exercise routine is assumed as a keystone in the management of sarcopenia. Nevertheless, currently there are no FDA-approved treatments for the management of sarcopenia. Human growth hormone and dehydroepiandrosterone are poorly effective or inactive. Although growth hormone increases protein synthesis and muscle mass, no improvements in strength and function are reported [11]. Testosterone or other anabolic steroids have also been questioned. Mild effects on muscle strength and mass can be observed, but their adverse effects limit their application, such as risk increase in male prostate cancer, female virilization, and cardiovascular pathology [11]. In this regard, there is a great interest in the development of herbal supplements for the treatment of sarcopenia, in particular for ameliorating muscle mass. A large number of herbal compounds have been reported to have effects on SKM but only a few of them showed modest effects in human studies [12]. Limited data support the use of human supplementation concerning the efficacy, safety, side effects, and adverse effects, and eventual pharmacological interactions. Going in this direction, i.e., for evaluating herbal compounds as a treatment for sarcopenia we analyzed hydrogen sulphide (H_2_S) and H_2_S-releasing molecules.

H_2_S is an endogenous gasotransmitter, the latest discovered, that can be found in several anatomical systems and tissues, including the SKM, in both animals and humans [13,14]. Current knowledge of H_2_S endogenous biosynthesis describes the involvement of four main enzymes, including cystathionine-γ-lyase (CSE), cystathionine-β-synthase (CBS), and cysteine aminotransferase (CAT), that operate in tandem with 3-mercaptopyruvate sulfurtransferase (3-MST) [15]. CBS and CSE both pyridoxal-5-phosphate-dependent enzymes utilize the amino acid L-cysteine as a substrate [16]. Contrariwise, 3-MST is a pyridoxal-5-phosphate-independent enzyme and uses 3-mercaptopyruvate as a substrate [17,18,19]. Several studies have proposed a physiological function for H_2_S in SKM wasting and homeostasis [13,14]. In our opinion, three key findings demonstrate the involvement of L-cysteine/CSE-CBS/H_2_S in the physiopathological mechanisms of SKM: (i) a low-cysteine diet regimen in CSE-deficient mice led to acute lethal myopathy [20]; (ii) hypercontractile SKM can be observed in patients prone to malignant hyperthermia, coupled to higher levels H_2_S and overexpression of CBS [19]; (iii) CBS-deficient mice with hyperhomocysteinemia showed an SKM dysfunction [21]. Overall, it would seem that low levels of H_2_S are associated with a scant SKM performance, whilst high levels are associated with hypercontractility/susceptibility. To support this, pieces of evidence have suggested that consumption of an antioxidant supplement containing cysteine may slow down or prevent age-related muscle loss [22]. Moreover, the Duchenne muscular dystrophy (DMD) mice model (mdx mice) has analogies to the CSE^−/−^ model fed with low-cysteine food, both displaying progressive muscle weakening and degeneration [23,24,25]. This evidence is reinforced by Terrill and colleagues, who observed a decrease in dystropathology and oxidative stress in L-cysteine-treated mdx mice [26]. More recent evidence demonstrates that, in myoblasts of DMD patients, there is an impairment in H_2_S signalling; indeed, a lowering in the gene expression of the CBS and CSE and H_2_S production was found [27]. Importantly, the exogenous replacement of H_2_S improves the molecular features of DMD in terms of inflammation and fibrosis in mdx mice [27].

In the present work, we investigated: (i) the involvement of H_2_S signalling in an in vitro model of sarcopenia based on damage induced by dexamethasone (DEX) on C2C12-derived myotubes (ii) the protective potential of an endogenous and exogenous source of H_2_S. For this purpose, glucoraphanin, a glucosinolate occurring exclusively in the botanical order Brassicales, as an exogenous source of H_2_S [28], L-cysteine, as an endogenous source of H_2_S, and 3-mercaptopyruvate, either as an exogenous or endogenous source of H_2_S [29] have been investigated.

## 2. Results

### 2.1. Impairment of H_2_S Signalling in DEX-Induced In Vitro Model of Sarcopenia in C2C12 Myotubes

DEX treatment (1 μM) for 48 h induces in vitro model of sarcopenia in C2C12-derived myotubes, as previously reported [30]. Here, we found that treatment with DEX significantly reduced the expression of both CBS and CSE in C2C12 myotubes (Figure 1A–C, * *p* < 0.05; ** *p* < 0.01). Furthermore, a reduction in H_2_S production was observed (Figure 1D). Indeed, the basal amount of H_2_S was significantly lowered in homogenates of DEX-treated C2C12 myotubes compared with vehicle-treated ones (** *p* < 0.001, Figure 1D).

### 2.2. C2C12 Myotube Viability and Caspase-3 Activity

C2C12 myotubes were cotreated for 48 h with DEX (1 μM) and with glucoraphanin (30 µM), L-cysteine (150 μM), and 3-mercaptopyruvate (150 μM) to assess their protective properties. As shown in Table 1, DEX (in absence or presence of tested compounds) did not modify cell viability. As shown in Figure 2, 48 h of DEX treatment (1 μM) induced apoptotic processes by promoting caspase-3 activity (increased by 70% in comparison with the control; ** *p* < 0.01). A 48 h cotreatment with glucoraphanin (30 µM) and mercaptopyruvate (150 µM) prevented the DEX-induced proapoptotic activity (^^ *p* < 0.01 versus DEX). L-cysteine 150 μM was not as protective.

### 2.3. Protective Effects from Oxidative Stress Induced by DEX

The treatment with DEX (1 µM) evoked oxidative imbalance, enhancing catalase activity (Figure 3) and increasing up to two folds both levels of O_2_^−^. (Figure 4) and protein carbonylation (Figure 5). As shown in Figure 3, glucoraphanin (30 μM), L-cysteine (150 μM), and mercaptopyruvate (150 μM) were effective in reducing catalase activity (^^ *p* < 0.01 versus DEX, for glucoraphanin; ^ *p* < 0.05 versus DEX for L-cysteine and mercaptopyruvate). The three compounds were also able to prevent the increase in O_2_^−^ levels (Figure 4); in particular, L-cysteine was able to bring the concentration back to control levels (^^ *p* < 0.01 versus DEX). As shown in Figure 5, protein carbonylation was significantly prevented by all of the three compounds (^ *p* < 0.05 versus DEX for L-cysteine; ^^ *p* < 0.01 versus DEX for glucoraphanin and mercaptopyruvate).

### 2.4. Morphology and Morphometrics of C2C12 Myotubes

C2C12 myoblasts completely differentiate in myotubes after 7 days of culture in a differentiation medium (DM). The following 48 h treatment with DEX (1 μM), altered the morphology and morphometry of myotubes (Figure 6A–E), reducing the diameter of myotube (Figure 6F) and the quantity of multinucleated myotubes (Figure 6G) by about 39% and 52%, respectively. The 48 h cotreatment with glucoraphanin (30 μM), L-cysteine (150 μM), and mercaptopyruvate (150 μM) prevented the morphological alterations (Figure 6A–G). The number of nuclei per myotube was not modified by either DEX or the three compounds (Figure 6H).

## 3. Discussion

Several mechanisms and risk factors, including poor physical activity, smoking, malnutrition, and age-related hormonal changes and cytokine levels, contribute to the development of sarcopenia [4,5]. The aforementioned mechanisms enclose changes in muscle protein turnover, muscle tissue remodelling, muscle cell recruitment, and apoptosis leading to muscle atrophy [9]. Sarcopenia is considered part of the aging process, so much so that it now represents a key point of research and public policy debate for its impact on morbidity, mortality, and healthcare costs. One of the major factors implied in the outbreak of sarcopenia is age-related insulin resistance [31]. Indeed, insulin is an anabolic hormone that not only promotes glucose absorption, glycogenesis, and lipogenesis but also stimulates skeletal muscle protein synthesis [32]. In line with this, the association between diabetes and sarcopenia and frailty is well proved by several reports [33]. Older diabetic patients have high incidence of frailty (32–48%) compared with nondiabetic older subjects (5–10%) [34,35,36]. To date, the cause/mechanism by which diabetes often coexists with sarcopenia/frailty remains to be clarified [37,38]. In a cohort of elderly patients suffering both hypertension and frailty, a correlation between physical decline and cognitive impairment was found. This evidence further stresses the role of comorbidities in the context of sarcopenia [39]; thus frailty represents an additional problem to be considered along with the control of blood pressure in older adults. More in general, frailty is frequently accompanied by comorbidities such as diabetes, heart failure, and hypertension in older subjects; therefore, a form of prevention of sarcopenia could result from better management of these populations of patients [40]. Furthermore, there are suggestions of a correlation between chronic kidney disease and muscle impairment, leading to a high incidence of frailty and an increased risk for mortality [41].

In addition, another critical cause implicated in the development of sarcopenia is the use of drugs such as chemotherapies and glucocorticoids [5,9,10]. In fact, a strong association between aging and sarcopenia development exists also following dexamethasone treatment [42]. Indeed, an excess of dexamethasone markedly reduces muscle strength and physical activity in animals [43] and humans [44]. This is not surprising considering that glucocorticoids can trigger muscular atrophy, by both inhibiting synthesis and accelerating degradation of muscle proteins in preclinical models, both in vivo and in vitro [45,46,47,48]. Noteworthy, sarcopenia is not only an age-related disease but it can also hit younger subjects affected by autoimmune diseases, such as rheumatoid arthritis [6]. At last, acute sarcopenia arises in survivors of COVID-19 too [49]. Thus, despite its clinical relevance, sarcopenia is still badly handled in the current clinical practice. In this scenario, H_2_S may offer new pharmacological approaches for the management of sarcopenia. Indeed, it is well accepted the role of H_2_S in the physiopathology of SKM [14,20,27] but also in the pathogenesis of diabetes [50] and not last as an important mediator of inflammation [51,52,53]. Considering this evidence, we assessed the efficacy of endogenous or exogenous sources of H_2_S in an in vitro model of sarcopenia. We evaluated the effects of the following in DEX-induced sarcopenia in C2C12-derived myotubes: glucoraphanin, a natural H_2_S donor; L-cysteine, the endogenous substrate for H_2_S production; and 3-mercaptoyruvate, an endogenous and exogenous source for H_2_S [29]. The potential efficacy of the abovementioned agents was strongly given by the finding that DEX-induced sarcopenia in C2C12 myotubes, both CBS and CSE expression was reduced, coupled with a significant reduction in H_2_S content. Thus, in our experimental condition, DEX negatively impacts the H_2_S biosynthesis in C2C12-derived myotubes. The fact that DEX reduces the CBS and CSE expression has also been observed in animal models such as DEX-induced hypertension and osteoporosis in the rat [54,55] or LPS-induced endotoxic shock [56]. In addition, we also found that in C2C12-derived myotubes DEX increased caspase 3 activity, the canonical biochemical indicator of both early and late phases of apoptosis [57] reinforcing the involvement of the apoptotic process in sarcopenia. Indeed, growing evidence suggests that muscle fibre atrophy is strongly correlated with an increase in muscle cell apoptosis [58,59]. Among the agents tested, glucoraphanin and mercaptopyruvate showed greater efficacy for reducing the increase in caspase 3 activity, while L-cysteine did not affect it. This apparent discrepancy could be related to a different kinetic in releasing H_2_S, based on the fact that either glucoraphanin or 3-mercaptopyruvate releases H_2_S very quickly, i.e., already within 20 min [29,60], while L-cysteine needs to be metabolized, requiring more time to release H_2_S. As reported by studies employing cell-free systems, cytochrome *c* is able to prime apoptosis-like shift in cytosols derived from several cell types [61,62,63,64]. In particular, cytochrome *c*, through caspase activation, speeds up apoptotic events in cell extracts, such as degradation of various caspase substrates and nuclear condensation and fragmentation. H_2_S can be placed in this context, acting as a cytochrome *c* inhibitor [65,66] and thus interfering with the cascade of caspases activation. Therefore, the timely release of H_2_S from glucoraphanin, L-cysteine, or 3-mercaptopyruvate could be crucial for inhibiting cytochrome *c* to block the caspase activation. The mitochondrial impairment leads to a redox unbalance [67], another important issue of sarcopenia; in our experimental condition, DEX caused an increase in both catalase activity and O^2−^ levels that resulted in reduced by glucoraphanin, 3-mercaptopyruvate, and L-cysteine. Protein carbonylation, the major hallmark of oxidative-stress-related disorders, also plays a role in labelling damaged proteins during oxidative stress to eliminate them from the biological system [68]. Moreover, we found that protein carbonylation was higher in presence of DEX and reduced by the treatment with all three H_2_S-releasing agents. This effect relies on the unquestionable antioxidant ability of H_2_S. Indeed, it is well known that H_2_S can readily scavenge reactive oxygen and nitrogen species, at higher rates than other canonical antioxidants such as glutathione [69]. The cytoprotective effects of the H_2_S-releasing molecules were also highlighted in myotubes. Our results show that DEX treatment leads to an impairment of morphology, decreasing the diameter and the formation of mature myotubes, whereas glucoraphanin, 3-mercaptopyruvate, and L-cysteine were able to restore the alterations in myotube diameter and number of multinucleated cells. It is well known that an oxidative environment negatively impacts muscle regenerative ability, this is the case of aging. Oxidative intracellular unbalance impairs the differentiation of myoblasts, while myogenesis is promoted by reducing conditions [70]. The same is confirmed also in oxidative conditions related to hyperglycemia [71]. The reason that the reduction in oxidative stress is correlated with the improvement of myotube formation is currently unclear, but in the literature, there is recent pieces of evidence linking calmodulin (CaM) to this phenomenon [72]. CaM is a fundamental regulator protein of muscle physiology, orchestrating functions such as cell proliferation, cell death, and muscle tissue remodelling [73]. CaM is rich in methionine (Met) and therefore particularly sensitive to oxidative triggers [74]. A single amino acid substitution of Met with Met sulfoxide (M109Q) in one or both alleles of the *CALM1* gene, which is one of three genes encoding the muscle regulatory protein CaM, strongly impaired C2C12 mouse myoblasts differentiation in myotubes [72]. Therefore, CaM seems to act a redox sensor, blocking myogenesis as consequence of oxidative stress [72].

Taking into account that the modification of the extracellular environment (i.e., the satellite cell niche) could improve the functionality of aged muscle precursor cells [67], our result hints to an important restorative effect elicited by H_2_S able to contrast the muscle atrophy. However, in this scenario, it is important to stress the role of H_2_S in diabetes, while considering the fact that SKM is an important regulator of glucose homeostasis in the human body and that sarcopenia is frequently accompanied by diabetes in older subjects. Indeed, plasma levels of L-cysteine, glutathione, and H_2_S are reduced in diabetic conditions [75,76]; as postulated by in vitro and in vivo studies, a potential connection exists in diabetes between the decreased levels of H_2_S and the impaired glucose homeostasis [77,78]. In addition, it has been reported that a diet with great consumption of organosulfur compounds, such as chives, leeks, garlic, and onions, participates in the recovery of circulating levels of H_2_S positively influencing the metabolic state [79,80]. In addition, as reported in an in vitro study performed on C2C12-derived myotubes, the deletion of CSE leads to a decrement in cellular glutathione concentration and makes the cells susceptible to oxidative stress raising ROS production, while augmentation of GSH levels and a reduction in cellular oxidative stress have been observed after supplementation with NaHS [75]. Similar results have been obtained by Sinha-Hikim and co-workers. They elegantly showed dietary supplementation of a cystine-ameliorated muscle structure and functionality in old mice [22].

In conclusion, this study provides molecular insights into the relevance of the CSE/CBS/H_2_S system in sarcopenia. H_2_S-releasing molecules, targeting diminishment of muscle cell apoptosis, reduction in oxidative stress, and control of glucose homeostasis, may represent a framework for therapeutic intervention in the management of sarcopenia. Furthermore, supplementation with H_2_S-releasing natural compounds, such as glucoraphanin, or alternatively a diet reach in Brassicales, may be also useful in the prevention of sarcopenia. The advantage of using glucoraphanin relies on its H_2_S slow-releasing properties and on its metabolization by intestinal microbiota, resulting in sulforaphane which, in its turn, exerts antioxidant, detoxifying, anti-inflammatory, and antiapoptotic effects. [81,82]. For these reasons, and based on our data, glucoraphanin stands out among other H_2_S donors and could be more effective than L-cysteine or 3-mercaptopyruvate in prevention and treatment of sarcopenia.

## 4. Materials and Methods

### 4.1. Cell Cultures

C2C12 mouse skeletal myoblasts were purchased from American Type Culture Collection (Manassas, VA, USA), cultured in Dulbecco’s Modified Eagle Medium (DMEM) supplemented with 10% foetal bovine serum (FBS), 100 U/mL penicillin, 100 µg/mL streptomycin, 200 mM L-glutamine (Life Technologies Italia, Milan, Italy) and maintained at 37 °C in a humidified atmosphere with 5% CO_2_. To differentiate myoblasts into myotubes, C2C12 myoblasts were cultured to reach 80% of confluence; cells were then shifted to DM (DM: DMEM supplemented with 2% horse serum, 100 U/mL penicillin, and 100 µg/mL streptomycin, 200 mM L-glutamine) for 7 days. DM replacement was performed every 2 days.

### 4.2. Pharmacological Treatments

After confluence attainment myoblasts were plated in proper cell culture plates (Corning Costar, Sigma-Aldrich, Milan, Italy) according to experimental procedures. Once differentiated in myotubes, they were treated with 1 µM DEX (Sigma Aldrich, Milan, Italy) for 48 h. Glucoraphanin (30 µM), purified from Tuscan black kale seeds by Bologna laboratory (CREA-AA; previously CRA-CIN) according to a previously described method [83], L-cysteine (150 µM) and 3-mercaptopyruvate (150 µM) (Sigma-Aldrich, Milan, Italy) were used in the presence of DEX. All treatments were done in DM. Concentration and time of exposure for DEX-induced damage were previously set up [30]; glucoraphanin concentration was chosen based on the ability of the compound to increase the intracellular H_2_S concentration [84]; L-cysteine and 3-mercaptopyruvate concentrations were chosen based on previously obtained results [29].

### 4.3. Cell Viability Assay (MTT Test)

Myotube viability was assessed by evaluating the reduction in 3-(4,5-dimethylthiazol-2-yl)-2,5-diphenyltetrazolium bromide (MTT) as an index of mitochondrial functionality. Cells were plated into 96-wells cell culture plates (3 × 10^3^ cells/well) and grown until 80% confluence. After 7 days of differentiation in DM, cells were treated with 1 µM DEX in absence or presence of glucoraphanin (30 µM), L-cysteine (150 µM), and 3-mercaptopyruvate (150 µM) for 48 h. After abundant washing, 1 mg/mL MTT was added to each well and incubated for 30 min at 37 °C. Upon the formation of formazan, 150 mL of dimethyl sulfoxide were added to each well to dissolve the crystals. The absorbance was measured at 550 nm. Experiments were performed in quadruplicate on at least three different experimental sets.

### 4.4. Western Blot Analysis

C2C12 myotube homogenates were prepared in modified RIPA buffer (50 mM Tris-HCl pH 8.0, 150 mM NaCl, 0.5% sodium deoxycholate, 0.1% sodium dodecyl sulphate, 1 mM EDTA, 1% Igepal) (Roche Applied Science, Italy) and protease inhibitor cocktail (Sigma-Aldrich, Milan, Italy). After the evaluation of protein concentration (Bradford assay, Sigma-Aldrich, Milan, Italy), denatured proteins were separated in 10% sodium dodecyl sulphate-polyacrylamide gels and then transferred to a poly-vinylidene fluoride membrane. Membranes were incubated with mouse monoclonal antibody for CSE (1:1000; Abnova, Milan, Italy) or rabbit polyclonal for CBS (1:1000; Santa Cruz Biotechnology, Heidelberg, Germany) and then with horseradish-peroxidase-conjugated secondary antibody, before the development with Chemidoc (Biorad, Milan, Italy). GAPDH (1:5000, Sigma-Aldrich, Milan, Italy) was considered as the housekeeping to normalize the protein band intensity [85]. Data were calculated as optical density (OD)*mm^2^.

### 4.5. H_2_S Determination

H_2_S production was measured in homogenates of C2C12 myotube treated with vehicle or DEX. Sample lysis was performed in a modified potassium phosphate buffer (100 mM, pH 7.4, sodium orthovanadate 10 mM, and proteases inhibitors) and Bradford assay was used to determine protein concentration. In order to evaluate the basal content of H_2_S, pyridoxal-5′-phosphate (2 mM) was added to the homogenates. After incubation at 37 °C for 40 min, trichloroacetic acid solution (TCA, 10% wt/vol), zinc acetate (1% wt/vol), N,N-dimethyl-p-phenylenediamine sulphate (DPD; 20 mM) in HCl (7.2 M) and FeCl_3_ (30 mM) in HCl (1.2 M) were added to each sample [86]. All samples were performed in duplicate, and H_2_S concentration was measured by optical absorbance at a wavelength of 668 nm and calculated against a standard curve of NaHS (3–250 μM). Data were calculated as nmol/mg protein*min^−1^.

### 4.6. Caspase-3 Activity

Myoblasts were plated into 6-well plates (5 × 10^4^ cells/well) and cultured upon reaching 80% confluence. After 7 days of differentiation in DM, myotubes were then incubated with DEX (1 µM) with or without glucoraphanin (30 µM), L-cysteine (150 µM), or 3-mercaptopyruvate (150 µM) for 48 h. Following the treatment, cell lysis was performed by adding 100 μL of lysis buffer (200 mM Tris-HCl buffer, pH 7.5, containing 2 M NaCl, 20 mM EDTA, and 0.2% Triton X-100) and centrifuged at 11,000× *g* for 10 min at 4 °C. A measure of 50 μL of the supernatant was collected and incubated with 25 μM fluorogenic peptide caspase substrate rhodamine 110 bis (N-CBZ-L-aspartyl-L-glutamyl-L-valyl-L-aspartic acid amide) (Molecular Probes, Milan, Italy) at 25 °C for 30 min. Cleaved substrate quantification for each sample was performed in a 96-well plate fluorescence spectrometer (FlexStation 3, Molecular Devices; excitation at 496 nm and emission at 520 nm). Caspase-3 activity for each sample was normalized to protein concentration, determined using bicinchoninic acid assay (Sigma Aldrich, Milan, Italy).

### 4.7. Catalase Activity

Myoblasts were plated into 6-well plates (5 × 10^4^ cells/well) and cultured upon reaching 80% confluence. After 7 days of differentiation in DM, myotubes were then incubated with DEX (1 µM) with or without glucoraphanin (30 µM), L-cysteine (150 µM), or 3-mercaptopyruvate (150 µM) for 48 h. Following the treatment, cell lysis was performed by adding 100 μL of lysis buffer (200 mM Tris-HCl buffer, pH 7.5, containing 2 M NaCl, 20 mM EDTA, and 0.2% Triton X-100) and centrifuging at 11,000× *g* for 10 min at 4 °C. The supernatant was collected and used to measure catalase activity by Amplex Red Catalase Assay Kit (Invitrogen, Monza, Italy) following the manufacturer’s instructions. Catalase activity for each sample was normalized to protein concentration, determined using bicinchoninic acid assay (Sigma Aldrich, Milan, Italy).

### 4.8. Superoxide Dismutase (SOD)-Inhibitable Superoxide Anion (O_2_^−^) Production Evaluated by Cytochrome c Assay

Myoblasts were plated in 6-well plates (5 × 10^4^ cells/well) and grown until 80% confluence. After 7 days of differentiation in DM, myotubes were then incubated with DEX (1 µM) in absence or presence of glucoraphanin (30 µM), L-cysteine (150 µM), or 3-mercaptopyruvate (150 µM) for 48 h. After treatment, cells were incubated in serum-free DMEM containing cytochrome *c* (1 mg/mL) for 4 h at 37 °C. Nonspecific cytochrome *c* reduction was evaluated by performing tests in the presence of bovine SOD (300 mU/mL). After collecting the supernatants, the optical density was spectrophotometrically measured at 550 nm. The SOD-inhibitable O_2_^−^ amount was calculated by subtracting nonspecific absorbance and by using an extinction coefficient of 2.1 × 10^4^ M^−1^ cm^−1^ and expressed as µM/mg protein/4 h. We chose incubation time based on preliminary experiments, which pointed out poor reliability after longer cellular environment exposure to cytochrome *c*.

### 4.9. Carbonylated Protein Evaluation

Myoblasts were plated in a 25 cm^2^ cell culture flask (7 × 10^4^ cells/flask) and cultured upon reaching 80% confluence. Cells were differentiated in myotubes in DM for 7 days. Carbonylated proteins were evaluated after 48 h incubation with 1 µM DEX, both alone and with glucoraphanin (30 µM), L-cysteine (150 µM) or 3-mercaptopyruvate (150 µM). After treatment and PBS washing, myotubes cell cultures were scraped on ice with lysis buffer containing 50 mM Tris-HCl pH 8.0, 150 mM NaCl, 1 mM EDTA, 0.5% Triton X-100, and complete protease inhibitor (Roche, Milan, Italy). Cell suspensions were collected, subjected to a freeze–thaw cycle, and centrifuged at 13,000× *g* for 10 min at 4 °C. Bicinchoninic acid assay was used to determine protein concentration. Twenty micrograms of proteins from each sample were denatured by 6% SDS and derivatized 10 mM 2,4-dinitrophenyl hydrazine (DNPH; Sigma-Aldrich, Milan, Italy) for 15 min at room temperature. Protein separation was carried out by using a 12% SDS–polyacrylamide gel by electrophoresis. Proteins were transferred onto nitrocellulose membranes (Bio-Rad, Milan, Italy). Membranes were incubated with blocking solution (PBS containing 0.1% Tween 20 (PBST) and 1% BSA; Sigma-Aldrich, Milan, Italy) and then overnight with specific primary antibody versus DNPH (Sigma-Aldrich) (1:5000 in PBST/1% BSA). After washing with PBST, the membranes were incubated for 1 h with horseradish-peroxidase-conjugated secondary anti-rabbit (1:5000 in PBST; Cell Signaling, Milan, Italy) and washed again. Peroxidase-coated bands were visualized with enhanced chemiluminescence (Pierce, USA). Scion Image analysis software was used to perform densitometric analysis. The density of all the bands for each condition was reported as the mean and normalized to β-Actin expression [87].

### 4.10. Morphologic Evaluations

C2C12 myoblasts were plated on coverslips (5 × 10^3^ cells/slice) and grown until 80% confluence. Cells were differentiated in myotubes in DM for 7 days. Differentiated myotubes were treated with 1 µM DEX in absence or presence of glucoraphanin (30 µM), L-cysteine (150 µM), and 3-mercaptopyruvate (150 µM) for 48 h. Afterward, cells were fixed and immunolabeled with either FITC-labelled phalloidin (1:40; Sigma-Aldrich, Milan, Italy) to stain filamentous actin and 4′,6-Diamidine-2′-phenylindole dihydrochloride (DAPI, 1:2000; Sigma-Aldrich, Milan, Italy) to stain the nuclei. The evaluation of myotubes diameter, number of nuclei for myotube, and multinucleated cells were performed using image analysing software (ImageJ 1.48). Morphometric analysis was carried out by collecting at least three independent fields through a 20 × 0.5 NA objective and micrographs to be analysed were taken using a motorized Leica DM6000B microscope equipped with a DFC350FX camera (Leica, Mannheim, Germany).

### 4.11. Statistical Analysis

Results were expressed as mean ± S.E.M. Analysis of variance (ANOVA) or Student’s t-test was performed as needed. A Bonferroni significant difference procedure was used as a post- hoc comparison. All assessments were made by researchers blinded to cell treatments. Data were analysed using the “Origin 8.1” software (OriginLab, Northampton, MA, USA) and GraphPad Software (Prism 8, San Diego, CA, USA). *p* < 0.05 was considered significant.

## Figures and Tables

**Figure 1 ijms-23-05955-f001:**
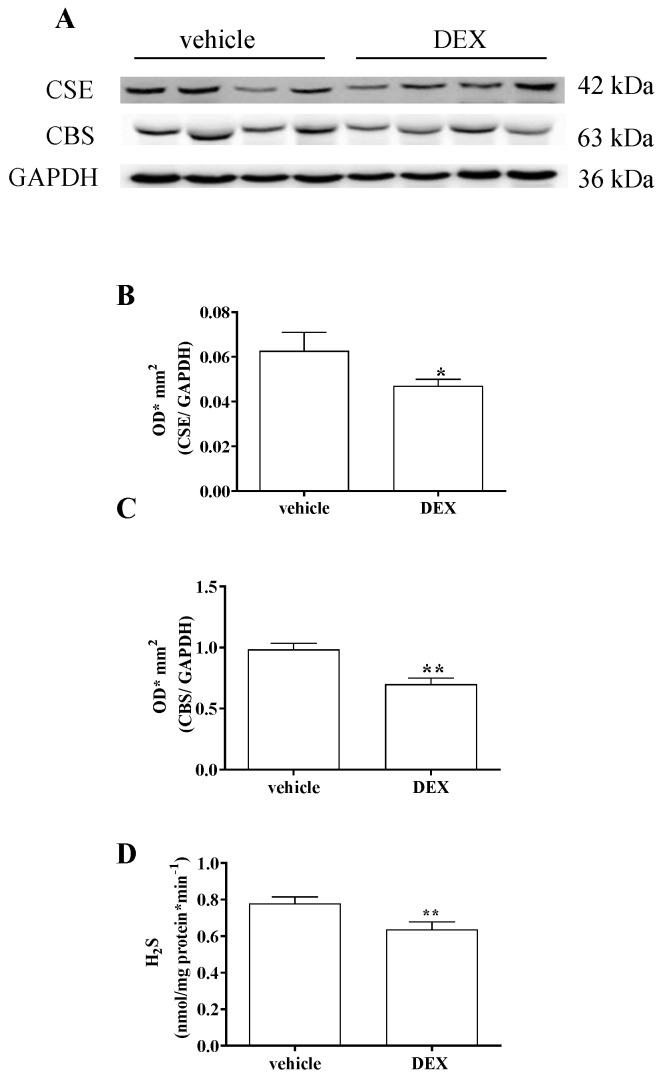
Illustrative Western blot for the expression of CBS and CSE in homogenates of C2C12 myotubes (**A**). CSE (**B**) and CBS (**C**) expression is significantly reduced in C2C12 myotubes treated with DEX compared with vehicle (* *p* < 0.05, ** *p* < 0.001). Results are normalized against a housekeeping protein, GAPDH. Data are expressed as optical density (OD)*mm^2^ and calculated as mean ± SEM of 5/6 experiments (*n* = 5 vehicle; *n* = 6 DEX). H_2_S production in vehicle-treated or DEX-treated C2C12 myotubes (**D**). The basal amount of H_2_S is significantly reduced in homogenates of C2C12 myotubes treated with DEX compared with vehicle (** *p* < 0.01). Results are expressed as nanomoles per milligram of protein per minute and values are calculated as mean ± SEM of 9 experiments.

**Figure 2 ijms-23-05955-f002:**
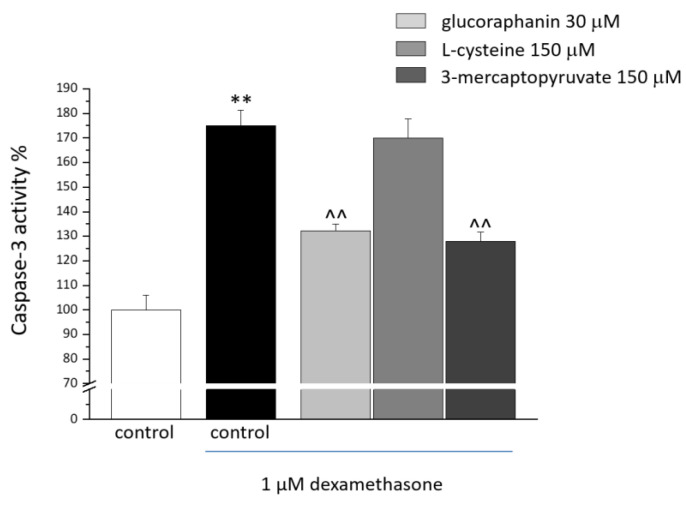
Caspase-3 activity in C2C12 myotubes. C2C12 myotubes were treated for 48 h with 1 μM DEX, both alone and in the presence of 30 μM glucoraphanin, 150 μM L-cysteine, and 150 μM mercaptopyruvate. Caspase-3 activity was quantified by a fluorescence assay. Values are expressed as % of activity and calculated as the mean ± S.E.M. of 3 experiments. ** *p* < 0.01 versus control; ^^ *p* < 0.01 versus DEX.

**Figure 3 ijms-23-05955-f003:**
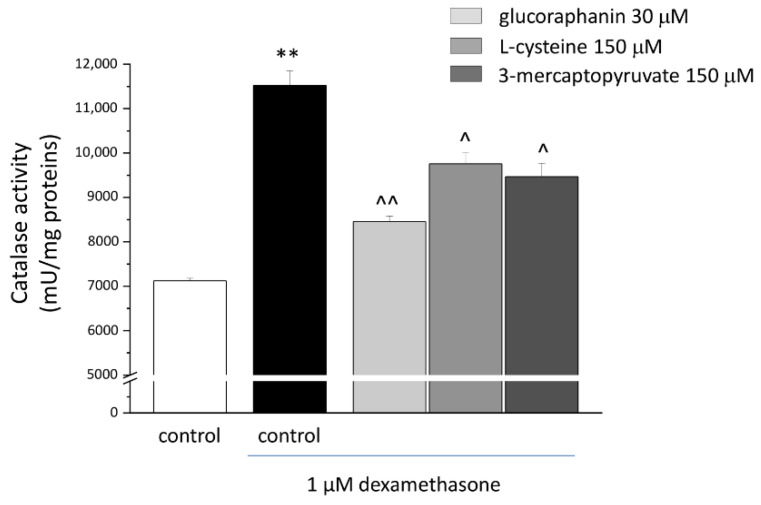
Catalase activity in C2C12 myotubes. C2C12 myotubes were treated for 48 h with 1 μM DEX, both alone and in the presence of 30 μM glucoraphanin, 150 μM L-cysteine, and 150 μM mercaptopyruvate. A fluorescence assay was used to quantify catalase activity. Values are expressed as mU/mg protein and calculated as the mean ± S.E.M. of 3 experiments. ** *p* < 0.01 versus control; ^ *p* < 0.05 and ^^ *p* < 0.01 versus DEX.

**Figure 4 ijms-23-05955-f004:**
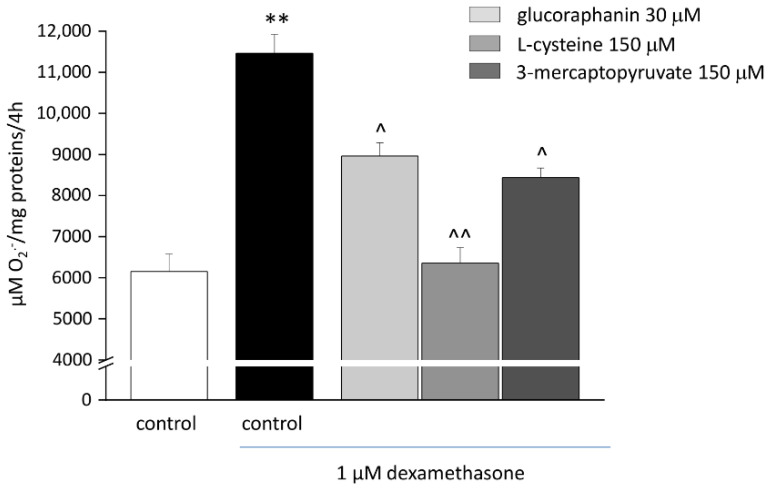
O_2_^−^. levels in C2C12 myotubes. C2C12 myotubes were treated with 1 μM DEX for 48, both alone and in the presence of 30 μM glucoraphanin, 150 μM L-cysteine, and 150 μM mercaptopyruvate. Cytochrome *c* assay was used to measure O_2_^−^. concentration. We measured the non-specific absorbance in the presence of superoxide dismutase (SOD) (300 mU/mL) and subtracted it from the total value. Values are expressed as μM/mg protein/4 h and calculated as mean ± S.E.M. of 3 experiments. ** *p* < 0.01 versus control; ^ *p* < 0.05 and ^^ *p* < 0.01 versus DEX.

**Figure 5 ijms-23-05955-f005:**
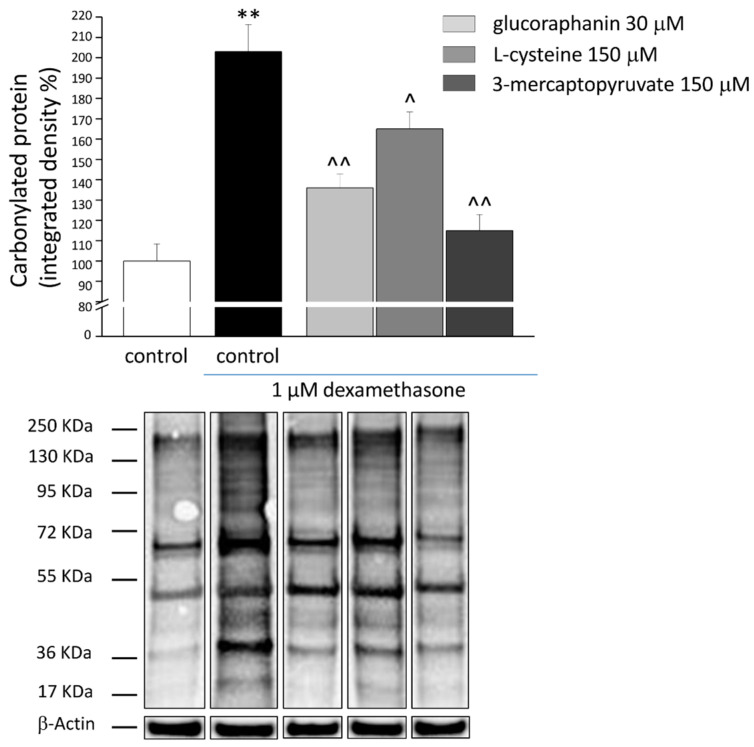
Protein carbonylation in C2C12 myotubes. C2C12 myotubes were treated for 48 h with 1 μM DEX, both alone and in the presence of 30 μM glucoraphanin, 150 μM L-cysteine and 150 μM mercaptopyruvate. Western blot analysis was carried out on cell homogenates using a specific antibody against DNPH. Figures show the densitometric analysis (**top**) and the illustrative immunoblot (**bottom**). Each sample was normalized to β-Actin expression. Values are expressed % of integrated density and calculated as mean ± S.E.M. of 3 different experiments. ** *p* < 0.01 versus control; ^ *p* < 0.05 and ^^ *p* < 0.01 versus DEX.

**Figure 6 ijms-23-05955-f006:**
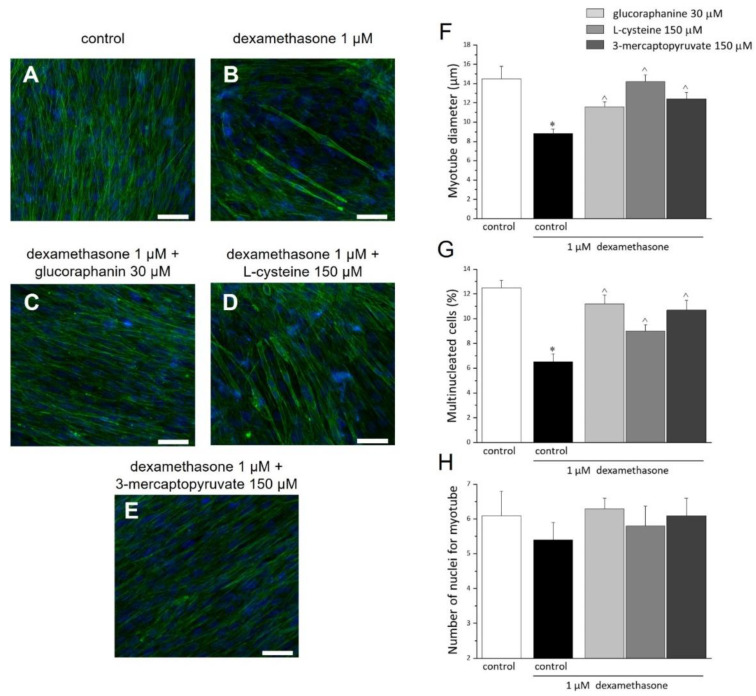
Morphological and morphometric analysis of C2C12 myotubes. C2C12 myoblasts were cultured in DM for 7 days. Once differentiated in myotubes, cells were incubated for 48 h with DM (**A**, control); with 1 μM DEX in the absence (**B**) or presence of 30 μM glucoraphanin (**C**), 150 μM L-cysteine (**D**), and 150 μM mercaptopyruvate (**E**). Cells were fixed and stained with FITC-conjugated phalloidin to highlight F-actin organization (green) and with DAPI, for nuclei staining (blue). Scale bar (white line): 50 μm. Morphometric analyses were carried out to provide: the length in μm of the myotube diameter (**F**); the multinucleated cells percentage (**G**) (percentage ratio of the number of multinucleated cells and the number of total cells, considered as 100%); and the number of nuclei in each myotube (**H**). The analyses were conducted in at least 3 random fields of each experimental group. Values are expressed as the mean ± S.E.M. of 3 different experiments. * *p* < 0.05 versus control; ^ *p* < 0.05 versus DEX.

**Table 1 ijms-23-05955-t001:** Cell viability in C2C12 myotubes (10 × 10^3^/well) treated for 48 h with DEX 1 μM in presence or absence of 30 μM glucoraphanin, 150 μM L-cysteine, and 150 μM mercaptopyruvate. Cell viability was evaluated by MTT assay. We arbitrarily set the control condition as 100%. Values are expressed as the mean ± S.E.M. of 3 experiments.

Treatment	Cell Viability
control	100 ± 2.4
dexamethasone 1 μM	92.3 ± 0.6
dexamethasone 1 μM + glucoraphanin 30 μM	103.3 ± 2.3
dexamethasone 1 μM + L-cysteine 150 μM	99.8 ± 3.1
dexamethasone 1 μM + 3-mercaptopyruvate 150 μM	105.4 ± 2.8

## Data Availability

Raw data are available upon request.

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
