# Peer review of "Beneficial Effect of H2S-Releasing Molecules in an In Vitro Model of Sarcopenia: Relevance of Glucoraphanin"

_ijms, 2022, doi:10.3390/ijms23115955_

Round 1

Reviewer 1 Report

Title: The title doesn't encapsulate the study well. In this study, three H2S-releasing compounds are used but the title only mentioned glucoraphanin. 

Results: The results seem incomplete. Why CSE and CBE expressions not studied for the treatment groups?

Why only one antioxidant enzyme was analysed?

It is uncertain how the suppression of oxidative stress leads to the improvement of myotubule formation. 

The discussion is presented as a single paragraph, which is not appropriate.

Author Response

Title: The title doesn't encapsulate the study well. In this study, three H2S-releasing compounds are used but the title only mentioned glucoraphanin.

We agree with the referee for the comment. We have better discussed the glucoraphanin effect in the conclusion. We believe now that the title is more appropriate.

Results: The results seem incomplete. Why CSE and CBE expressions not studied for the treatment groups?

We thank the referee for the comment. In this study our purpose was to investigate the beneficial effects of H2S-releasing molecules in a vitro model of sarcopenia encouraged also by the obtained results for what concern H2S signaling in terms of reduction in CBS/CSE expression and H2S production after dexamethasone treatment.  We focused on the ability of molecules (exogenous and endogenous sources of H2S) in restoring the dexamethasone-induced damage in C2C12 derived myotubes. The evaluation of the recovery of H2S signaling after the treatments with” H2S donors” was not in our pursuit. We looked at a more concrete possibility to use, in a future, these molecules to treat sarcopenia and in particular glucoraphanin, being a natural compound, and thus we tested their efficacy.

Why only one antioxidant enzyme was analysed?

We perfectly agree with the referee, the measurements related to redox balance could be extended to several others parameters. Nevertheless at this point of the research we have not yet hypothesized a signal of the redox machinery that could be candidated as pivotal in H2S protective effects. So we explored 3 different oxidative steps: an antioxidant enzyme like catalase, the level of ROS and consequent damage to proteins (carbonylation). It is uncertain how the suppression of oxidative stress leads to the improvement of myotubule formation. This is a pivotal point. It is well known that an oxidative environment negatively impacts muscle regenerative ability, this is the case of aging. Oxidative intracellular unbalance impairs myoblast differentiation, while reducing environments favor myogenesis (Hanson et al., 2007; doi: 10.1016/j.cellbi.2006.11.027). The same is confirmed also in oxidative conditions related to hyperglicemia ( Liu et al., 2020; https://doi.org/10.1016/j.yexcr.2020.112234). The reason why the reduction of oxidative stress leads to the improvement of myotube formation is currently unclear. Recently, calmodulin redox sensor alterations was linked to the phenomenon. A single amino acid substitution M109Q that mimics oxidation of methionine to methionine sulfoxide in one or both alleles of the CALM1 gene, one of three genes encoding the muscle regulatory protein calmodulin, in C2C12 mouse myoblasts strongly impaired differentiation in myotubes (Steil et al., 2020; doi.org/10.1371/journal.pone.0239047). Discussion was improved accordingly.

The discussion is presented as a single paragraph, which is not appropriate.

Concerning the discussion section, the division in paragraphs is not a Journal’s requirement.

Reviewer 2 Report

This is an interesting study exploring an in vitro model of sarcopenia. I would suggest you to revise the Manuscript since there are some misspellings. Also, please improve the English form. The quality of the figures should be increased. In rhe discussion section I would recommend you to speculate the role of comorbidities in sarcopenia. Please find some updated reference to cite and discuss:

Physical decline and cognitive impairment in frail hypertensive elders during COVID-19. Mone P, Pansini A, Frullone S, de Donato A, Buonincontri V, De Blasiis P, Marro A, Morgante M, De Luca A, Santulli G.Eur J Intern Med. 2022 May;99:89-92. doi: 10.1016/j.ejim.2022.03.012. Epub 2022 Mar 14.

Hypertension, heart failure, and frailty in older people: A common but unclear situation.Camafort M, Kario K.J Clin Hypertens (Greenwich). 2020 Oct;22(10):1763-1768. doi: 10.1111/jch.14004. Epub 2020 Aug 20.

Exercise and CKD: Skeletal Muscle Dysfunction and Practical Application of Exercise to Prevent and Treat Physical Impairments in CKD. Roshanravan B, Gamboa J, Wilund K.Am J Kidney Dis. 2017 Jun;69(6):837-852. doi: 10.1053/j.ajkd.2017.01.051. Epub 2017 Apr 18.

Author Response

This is an interesting study exploring an in vitro model of sarcopenia. I would suggest you to revise the Manuscript since there are some misspellings. Also, please improve the English form. The quality of the figures should be increased. In the discussion section I would recommend you to speculate the role of comorbidities in sarcopenia. Please find some updated reference to cite and discuss:

Physical decline and cognitive impairment in frail hypertensive elders during COVID-19. Mone P, Pansini A, Frullone S, de Donato A, Buonincontri V, De Blasiis P, Marro A, Morgante M, De Luca A, Santulli G.Eur J Intern Med. 2022 May;99:89-92. doi: 10.1016/j.ejim.2022.03.012. Epub 2022 Mar 14.

Hypertension, heart failure, and frailty in older people: A common but unclear situation.Camafort M, Kario K.J Clin Hypertens (Greenwich). 2020 Oct;22(10):1763-1768. doi: 10.1111/jch.14004. Epub 2020 Aug 20.

Exercise and CKD: Skeletal Muscle Dysfunction and Practical Application of Exercise to Prevent and Treat Physical Impairments in CKD. Roshanravan B, Gamboa J, Wilund K.Am J Kidney Dis. 2017 Jun;69(6):837-852. doi: 10.1053/j.ajkd.2017.01.051. Epub 2017 Apr 18.

We thank the referee for the appreciation of our results; we have addressed all the suggestions regarding the English form of the manuscript. We have improved the quality of the figures and we have discussed the role of comorbidities in sarcopenia, citing the references suggested.

Round 2

Reviewer 1 Report

- "after orally administration, glucoraphanin can be metabolized by the intestinal microbiota into sulfurane" - in that case, the authors should have used sulfurane in the study instead of glucoraphanin. What is the percentage of glucoraphanin absorbed into the blood stream unchanged?

- there are some grammatical errors throughout the manuscript which need to be corrected. exp: CaM is reach (rich?) in metionine (Met); after orally (oral?) administration

Author Response

Comments and Suggestions for Authors

- "after orally administration, glucoraphanin can be metabolized by the intestinal microbiota into sulfurane" - in that case, the authors should have used sulfurane in the study instead of glucoraphanin. What is the percentage of glucoraphanin absorbed into the blood stream unchanged?

We thank the referee for the comment. In the conclusion, we have stressed the beneficial effect of glucoraphanin compared to other agents tested on the basis of the evidence that glucoraphanin acts not only as a precursor to sulforaphane but also as a slow-release H2S molecule, according to recently reported data (Lucarini E et al; Phytother Res. 2018;32(11):2226-2234; Gambari L et al., l Cell. Nutrients. 2022; 14(3):435). Here we used glucoraphanin instead of sulforaphane precisely to demonstrate the contribution of H2S in improving DEX-induced damage in C2C12-derived myotubes.

For what concerns the query raised by the referee, Cwik et al (J Pharm Biomed Anal. 2010;52(4):544-9) have measured the glucoraphanin plasma levels in dogs and rats following the assumption of vehicle or glucoraphanin at the dose of 10, 50, 100, and 500 mg/kg/die for 13 days. They reported that “Glucoraphanin was not detected in the plasma of the control group. All animals receiving glucoraphanin had measurable levels of glucoraphanin in plasma. The mean of measured concentrations for groups dosed at 10, 50, 100, and 500 mg/kg/day were 49.9 ng/mL (range, 13.2 to 83.4 ng/mL); 198 ng/mL (range, 111 to 300 ng/mL); 416 ng/mL (range, 254 to 577 g/mL); and 1630 ng/mL (range, 1110 to 2470 ng/mL), respectively.”

Based on the growing interest in H2S-releasing compounds and their potentiality as promising drugs, several H2S donors have been identified and extensively studied. The major limitation for many donors is represented by their instability and rapid degradability. Therefore, taking into account Cwik’s and our data, and considering that the in vivo activity of glucoraphanin can be associated with both glucoraphanin itself and its metabolization by the gut microbiota to sulfurane, glucoraphanin stands out among others H2S donors

Reviewer 2 Report

The authors have address my concerns.

Author Response

We would like to thank the reviewer for the revision work and for the ultimate approval of the text.